# Age and Task Modulate Olfactory Sensitivity in the Florida Carpenter Ant *Camponotus floridanus*

**DOI:** 10.3390/insects14090724

**Published:** 2023-08-23

**Authors:** Stephen T. Ferguson, Isaac Bakis, Nicholas D. Edwards, Laurence J. Zwiebel

**Affiliations:** Department of Biological Sciences, Vanderbilt University, Nashville, TN 37235, USA; stephen.t.ferguson@vanderbilt.edu (S.T.F.); isaac.bakis@vanderbilt.edu (I.B.); nicholas.d.edwards@vanderbilt.edu (N.D.E.)

**Keywords:** hymenoptera, olfaction, odor coding, task allocation, age polyethism, aging

## Abstract

**Simple Summary:**

Florida carpenter ants (*Camponotus floridanus*) live in colonies comprised of thousands of workers. The smallest workers, known as minors, engage in routine tasks such as nursing and foraging while the largest workers, known as majors, are thought to be soldiers specialized for defending the nest. How ant colonies allocate their workforce to address the dynamic and ever-changing needs of the colonies remains an open question in the field, but current evidence suggests that ant social behavior likely results from a combination of genetic/epigenetic, physiological, and systems-level processes. Here, we extend these studies by investigating the role of olfactory sensitivity in regulating ant behavior. Minor workers exhibited significant shifts in olfactory sensitivity and odor coding as they aged and switched tasks. The olfactory sensitivity of majors, however, remained relatively stable as they aged. From these studies, we also identified a single compound, 3-methylindole, which elicited significantly higher olfactory responses and aversive behavior in nurses compared to foragers, suggesting that this chemical may have a role in brood care. Overall, these studies support the hypothesis that changes in olfactory sensitivity play an important role in regulating social behavior in ants.

**Abstract:**

Age-related changes in behavior and sensory perception have been observed in a wide variety of animal species. In ants and other eusocial insects, workers often progress through an ordered sequence of olfactory-driven behavioral tasks. Notably, these behaviors are plastic, and workers adapt and rapidly switch tasks in response to changing environmental conditions. In the Florida carpenter ant, smaller minors typically perform most of the work needed to maintain the colony, while the larger majors are specialized for nest defense and rarely engage in these routine tasks. Here, we investigate the effects of age and task group on olfactory responses to a series of odorant blends in minor and major worker castes. Consistent with their respective roles within the colony, we observed significant age-associated shifts in the olfactory responses of minors as they transitioned between behavioral states, whereas the responses of majors remained consistently low regardless of age. Furthermore, we have identified a unitary compound, 3-methylindole, which elicited significantly higher responses and behavioral aversion in minor nurses than in similarly aged foragers suggesting that this compound may play an important role in brood care. Taken together, our results suggest that age- and task-associated shifts in olfactory physiology may play a critical role in the social organization of ant colonies.

## 1. Introduction

Animals exhibit a remarkable diversity of behaviors and related sensory capabilities that are essential for their survival and reproductive success. In ants and other eusocial insects, workers often exhibit age polyethism, in which they progress through an ordered sequence of behaviors including nursing, midden work, and foraging [1,2]. Furthermore, in many ant species, olfaction is critical for social communication and organization [3] and plays a key role in mediating group behaviors such as brood care [4], foraging [5,6], and nestmate recognition [7,8]. In the formicine carpenter ant *Camponotus floridanus*, these tasks are further allocated between two morphologically distinct worker castes—smaller minors and larger majors. While minors perform most of the work necessary for colony survival, majors are less involved in routine tasks and instead appear to be specialized for nest defense, as soldiers [9,10,11]. Importantly, these distinct behavioral roles are also associated with differences in olfactory sensitivity and odor coding [11], which may be further modulated by integrative processing in the central nervous system [12,13].

Allocating a colony’s workforce to dynamically address diverse requirements for group survival is a fundamental process that underlies ant eusocial organization [14]. This division of labor is often tightly associated with age. Young, callow workers often remain inside the safety of the nest where they attend to the brood as nurses. Older, mature workers transition to more dangerous activities outside the nest, such as foraging and territorial defense [2,15]. While age polyethism has been observed in ants and other eusocial Hymenoptera [2,15,16,17,18], ant behavior is remarkably plastic in this regard, and workers are capable of rapidly switching tasks in response to real-time shifts in colony need [2,19]. For example, increasing the amount of debris near the nest or providing extra food to collect results in task switching as workers adjust their workforce to clean or forage, respectively [19]. In *C. floridanus*, experimental colonies comprised of only foragers resulted in random, age-independent task switching as workers reverted back to nursing; in colonies comprised of only nurses, however, the oldest nurses were significantly more likely to switch to foraging [2]. These results suggest that, in *C. floridanus*, task allocation is influenced by both age and previous experience; in addition, older workers may be more amenable to task switching than younger workers. In ants, it seems likely that age polyethism coupled with dynamic and flexible task switching facilitate efficient resource utilization through rapid adaptation to fluctuations in resource availability and other environmental disturbances, such as the loss of colony members.

Unraveling the mechanisms underlying age polyethism as well as age-independent task switching will contribute to our understanding of the intricate social dynamics and adaptive strategies employed by ants and other social animals. Several hypotheses, none of which are mutually exclusive, have been advanced to explain these processes. For example, in honeybees, worker tasks appear to be influenced, in part, by the transcript abundance of the foraging gene, which encodes a cGMP-dependent protein kinase (PKG) and is expressed at higher levels in honeybee foragers than in nurses [20,21,22]. Inexplicably, the opposite appears to be true in ants, where nurses and other workers within the nest exhibit higher levels of foraging transcripts than the forager workers [23,24]. Here it seems likely that foraging gene expression is more closely associated with age rather than task and may therefore only indirectly influence worker behavior [25]. Not surprisingly, insect hormone titers may also play a key role in task allocation. In the harvester ant *Pogonomyrmex californicus*, foragers had significantly higher levels of juvenile hormone (JH) and lower levels of ecdysteroids than ants performing tasks within the nest [26]. Similar trends have been observed in *C. floridanus*, where the neuronal corepressor CoREST acts to maintain higher levels of JH in foraging minors than in non-foraging majors [10]. Epigenetic modifications have also been shown to affect task allocation and worker behavior; in vivo treatment with histone deacetylase (HDAC) inhibitors induces minor-like foraging behavior in *C. floridanus* majors [9]. Task allocation in social insects can also be understood as a complex system, whereby collective behaviors emerge from self-organizing dynamics within the colony superorganism [27]. Through relatively simple local social interactions and feedback mechanisms, colony-level patterns of behavior such as foraging can arise within ant colonies [28]. 

Chemosensory-based communication and olfactory signaling in particular play central roles in mediating and, indeed, driving social behaviors of ants both within and outside of the colony [29]. At the molecular level, olfaction in ants and other insects involves the detection and processing of chemical cues through a complex interplay of extra- and intra-cellular processes [30]. Environmental odors initially enter as unitary signals or, more likely, as complex odorant blends through a diverse range of porous hair-like structures known as chemosensory sensilla, which are distributed along the antennae and other sensory appendages. Once signals enter the aqueous sensillar lymph, they encounter and interface with a range of soluble odorant binding protein chaperones and odor-degrading enzymes before they are able to bind and activate a wide range of odorant receptors (ORs), ionotropic receptors, and/or gustatory receptors located on the dendrites of olfactory sensory neurons (OSNs) [31]. The depolarization of these OSNs leads to action potentials that relay olfactory information first to the antennal lobe glomeruli and then to higher-order processing centers of the brain, including the mushroom bodies and lateral horn [32,33,34]. Relative to solitary insects, the OR gene family in ants has undergone a massive expansion [35,36,37,38,39,40], providing ants with the capacity to communicate and detect a wide range of social information, including cuticular hydrocarbons (CHCs) that convey caste and colony-specific information, and which may have facilitated their evolutionary transition to eusociality [41,42]. 

Understanding the interplay of genetics, epigenetics, physiology, and higher-order group dynamics is crucial for understanding how ants coordinate their collective behaviors and olfactory sensitivity to respond to the fluctuating demands of the colony. Here, we continue to explore these questions by conducting an electrophysiological response screen complemented by a behavioral valence bioassay to examine the hypothesis that a variation in olfactory sensitivity and odor coding is associated with age- and task-related changes in worker behavior in the carpenter ant *C. floridanus*. Consistent with this hypothesis, we found that minor workers displayed profound age- and task-associated shifts in olfactory sensitivity and odor coding that aligned with their more active and behaviorally flexible role within the colony. The seemingly less-active, yet more-specialized soldier majors, however, consistently displayed low-level responses to odors over time.

## 2. Materials and Methods

### 2.1. Animal Husbandry

Six laboratory-reared colonies of *C. floridanus* (Buckley 1866)—originating from field collections by Dr. J. Liebig (Arizona State University) from the Sugarloaf Key (D601) and Dr. S.L. Berger (University of Pennsylvania) from the Fiesta Key (C6, K17, K19, K34 and K39) in South Florida, USA—were maintained in individual container colonies in an incubation chamber with a 12 h light:12 h dark photoperiod at 25 °C and an ambient humidity of approximately 70%. Each colony was provided with a Bhatkar diet, crickets, 10% sucrose solution, and distilled water three times per week. Adult minor and major workers were used for all experiments.

### 2.2. Paint Marking, Behavioral Monitoring, and Collections

Callow minor and major workers in approximately equal numbers were identified and collected based on their soft, light-colored cuticle, low mobility, and proximity to the brood pile. Ants with these characteristics were likely to be less than 24 h post-eclosion at the time of collection. These callow workers were anesthetized briefly with CO_2_, and Sharpie™ oil-based paint pens were used to mark the head, thorax, and gaster with a unique, painted color code. Prior to making behavioral observations, colonies were removed from the incubation chamber for at least 5 min to allow them to acclimate after having been handled. If the colony was disturbed, for example when removing trash or replacing food, no observations were made that day. Over the course of about 1 year, the behavior of age-known ants engaged in pre-specified social behaviors—nursing (carrying brood), performing trophallaxis, cleaning (carrying trash), and foraging (eating crickets, eating Bhatkar diet, and drinking sugar water)—that could be identified based on their paint code was recorded as observed (N = 212).

### 2.3. Electroantennography

Electroantennogram (EAG) data were derived from a previously published report [11] and reanalyzed after the addition of age-related parameters. Briefly, the odor blends tested were comprised of 390 unitary compounds representing 11 different chemical classes: alcohols (Blends 1–6), aldehydes (7–9), alkanes (10), amines (11–14), carboxylic acids (15–18), esters (19–25), ketones/indoles (26–30), lactones (31–32), sulfurs (33–34), and thiazoles (35–36). The unitary odorants were selected to encompass a broad range of putatively biologically salient chemical cues, which were then organized into blends to facilitate high-throughput electrophysiological odor screening; a detailed list of these odorants can be found in Appendix A. 

EAGs were conducted using an IDAC-232 controller (Ockenfels Syntech GmbH, Buchenbach, Germany) and data were processed using EAG2000 software (Ockenfels Syntech GmbH). Odor stimuli were delivered using a Syntech CS-05 stimulus flow controller (flow rate of 1.5 cm^3^ s^−1^; Ockenfels Syntech GmbH). The heterocyclic compound 5,6,7,8-tetrahydroquinoline (TETQ) was diluted in diethyl ether to 10^−1^ M for use as a positive control to ensure the integrity of the biological preparation for each EAG experimental setup. Prior to all experimental EAG recordings, each preparation was first stimulated with the headspace of 10 µL diethyl ether, then TETQ (10^−1^ M), and then another control stimulus of diethyl ether alone. Normalization of the TETQ response was subsequently done through linear interpolation vis-à-vis EAG2000. Before every subsequent EAG recording, a TETQ response of at least 1.5 × diethyl ether solvent was required to ensure the consistency and integrity of each preparation, as well as to serve as a positive control [11]. All other odor blends and unitary compounds were dissolved in ND96 buffer (96 mM NaCl, 2 mM KCl, 1 mM MgCl_2_, 1 mM CaCl_2_, and 5 mM HEPES, pH 7.5) to a final concentration of 10^−3^ M, and all odor cartridges were filled with 10 µL of solution. Solvent (diethyl ether or ND96) normalization was performed using linear interpolation and then the solvent was subtracted from the response such that the normalized solvent response was set to 0. A detailed description of the methods can be found in [11].

### 2.4. Colony-Level Responses to 3-Methylindole

To assay colony-level responses to 3-methylindole (3MI), small pieces of filter paper (2.5 cm in diameter) (VWR, Whatman VWR, West Chester PA, USA) were soaked in 50 µL of solvent (DMSO) or serial dilutions of 3MI (10^−5^, 10^−3^, or 10^−1^ M). These were then distributed randomly throughout the colony. Each colony contained approximately 2000–5000 workers, a single queen, and numerous broods including eggs, larvae, and pupae in a plastic container (40.6 × 26.7 cm). The full colony was recorded for 10 min using a digital high-definition camera (Model HC-V750, Panasonic Corporation, Newark, NJ, USA) (Appendix A). Videos were then analyzed post hoc by counting the number of worker ants on each piece of paper every 30 s for 10 min, and the mean number of ants on each respective piece of filter paper was then calculated.

### 2.5. Individual Responses to 3-Methylindole

To assay individual responses, nurses (carrying brood) and foragers (consuming crickets, Bhatkar diet, or sugar water) were collected and placed into modified Petri dish arenas (150 mm in diameter) (Appendix A). The lid of these arenas had a single, small hole (1 cm in diameter) near the edge of the lid. A small square of mesh secured with double-sided sticky tape was placed over the top of this hole to allow ventilation but prevent escape from the arena. Prior to the start of the bioassay, a single ant was placed underneath a small lid (35 mm in diameter) in the Petri dish arena and allowed to acclimate for 10 min. After this acclimation period, but before releasing the ant, a small piece of filter paper (2.5 cm in diameter) (VWR, Whatman VWR, West Chester, PA, USA) soaked in either solvent (DMSO) or a serial dilution of 3MI (10^−5^, 10^−3^, or 10^−1^ M) was introduced into the arena underneath the ventilation hole. The small lid securing the ant during acclimation was then removed, and the ant was allowed to wander freely in the arena. The location of the ant was then digitally recorded for 10 min (Model HC-V750 Panasonic Corporation, Newark NJ, USA), and these videos were analyzed using an automated tracking software package (EthoVision^®^ XT v8.5, Noldus Information Technology, Wageningen, The Netherlands) to calculate proximity to the Whatman paper zone (cm) and the percentage of time spent standing directly on the Whatman paper zone. Proximity to the Whatman paper zone was then normalized by subtracting the mean distance ants spent in proximity of the solvent-alone treatment from their mean distance to 3MI-impregnated filter paper. This normalization resulted in a distance (in cm) of 0 relative to solvent alone. Positive normalized distance values indicate attraction and negative normalized distance values indicate repulsion (Appendix A).

### 2.6. Statistical Data Analysis

Statistical data analysis was performed using GraphPad Prism v9.1.2 (GraphPad Software, Inc., Boston, MA, USA). QQ and residual plots were used to assess the normality and variance of the residuals, respectively, before conducting statistical tests including One-Way and Two-Way ANOVAs and One Sample t-Tests. Data for Figures 2a and 3 were transformed to achieve normality and equal variance prior to conducting their respective Two-Way ANOVAs. Hierarchical clustering using a dendrogram was conducted using Plotly (Plotly Technologies Inc., Montreal, QC, Canada). Figures were created using Plotly (Plotly Technologies Inc.) and GraphPad Prism v9.1.2 (GraphPad Software, Inc., Boston, MA, USA).

## 3. Results

### 3.1. Morphological Castes Perform Different Tasks

Previous studies in *C. floridanus* have demonstrated that minor workers exhibit an age-associated transition from nursing to foraging and perform the majority of tasks that make up the maintenance work within the colony; in contrast, majors are generally less active and instead act as a specialized soldier caste within the colony [2,9,10,11]. To complement these studies and provide additional validation for this phenomenon within our laboratory-reared *C. floridanus* colonies, we tracked the age and behavior of adult workers within the nest. As expected, based on similar observations across many eusocial Hymenopteran, *C. floridanus* minor workers exhibited an age-associated transition from nursing (mean ± SEM = 28.11 ± 2.24 days post-eclosion, N = 101) to foraging (51.69 ± 3.41 days, N = 66) (Appendix A). Minors performing trophallaxis with nestmates were of intermediate age (31.09 ± 4.65 days, N = 22). Callow minors only engaged in nursing, and day 6 was the earliest that foraging behaviors were observed. After this timepoint, however, the behavior of minor workers appeared more flexible, with workers of all ages observed nursing, exchanging food, or foraging. In contrast to their relatively active minor siblings, major workers rarely engaged in foraging and indeed were never observed carrying brood or other behaviors associated with nursing (Appendix A). Those few majors that were engaged in trophallaxis (25.80 ± 5.83, N = 5) and foraging (48.30 ± 13.51, N = 10) were of similar age to their minor counterparts. Based on these observations (Appendix A), and for the purposes of this study, we elected to organize our experiments into two age bins across two morphological castes: callow minors and callow majors (<6 days post-eclosion) and mature minors and mature majors (≥6 days post-eclosion).

### 3.2. Age-Associated Shifts in Odor Coding Observed in Minors but Not Majors

To investigate the relationship between age and olfactory sensitivity in minors and majors, we analyzed EAG data across callow and mature age-groups, revealing antennal responses to a diverse collection of 36 odor blends. Hierarchical clustering of the untransformed EAG responses (i.e., mV) of age-known minors and majors revealed that worker responses can be broadly clustered into three groups, or clades: C_1_ (green), C_2_ (red), and C_3_ (blue/cyan/purple) (Figure 1). Of these, callow minors, callow majors, and mature majors largely clustered together in clades C_1_ and C_2_, where they comprised 84.0% and 64.2% of all observations, respectively. By contrast, C_3_ is almost exclusively comprised of mature minors (90.9%) (Figure 1). These clustering patterns suggest that there are fundamental odor-coding differences that distinguish behaviorally plastic, mature minors from the remaining work force in the colony. Indeed, mature minors exhibited profoundly higher EAG responses to every odor blend tested than callow minors, callow majors, and mature majors, and there was a significant effect of caste group on olfactory sensitivity (Two-Way ANOVA, *p*-value for caste < 0.0001; odor blend = 0.7459; caste × odor blend > 0.9999; Figure 2a). Moreover, the mean response to these odor blends was significantly higher in mature minors than in all other groups (One-Way ANOVA with Tukey’s multiple comparison test, *p*-value < 0.0001; Figure 2b). In addition, while the effect size was smaller, the mean response of callow minors to odor blends was also significantly lower than in callow majors and mature majors (*p* < 0.0001; Figure 2b). In contrast to the significant increase in olfactory sensitivity in aging minors, there was no significant difference in the mean EAG responses of callow and mature majors (*p* = 0.9007). An examination of representative EAG traces illustrates the profound shift in olfactory sensitivity and odor coding observed in mature minors when compared with other worker age groups within the *C. floridanus* colony (Figure 2c–f).

### 3.3. Task-Associated Shifts in Olfactory Sensitivity

Thus far, we have described the raw EAG responses of minors and majors as they age (Figure 2). While these results shed light on the distinctive differences in baseline olfactory sensitivity and odor coding, they do not control for the solvent-alone responses that were also notably elevated in mature minors when compared with the other worker groups (Figure 2a,d). After normalizing the data to the ND96 solvent-alone responses, significant, albeit more nuanced, differences were again observed in olfactory sensitivity with respect to age and odor blend (Two-Way ANOVA, *p*-value for caste < 0.0001; odor blend < 0.0001; caste × odor blend = 0.1426; Figure 3). While the EAG responses between callow and mature workers within each caste (e.g., callow minors vs. mature minors) were not significantly different (Tukey’s multiple comparison post-hoc test, *p*-value for minors = 0.2997; majors = 0.9973), every other pairwise comparison (e.g., mature minors vs. mature majors) was significant (*p* < 0.0001).

Our analysis focused next on the ketones and indoles that make up Blend 26, which collectively elicited the highest response in callow minors and indeed the highest response to any odor blend for all caste/age groups (Figure 3), suggesting that some of these compounds may be associated with the nursing behaviors most commonly observed in younger minors (Appendix A). The unitary compounds that comprise this blend include 1,3-diphenylacetone (13D), 2-octanone (2OCT), 3-acetyl-2,5-dimethylfuran (3A25), 3-methylindole (3MI), 3-octanone (3OCT), 4-(4-methoxyphenyl)-2-butanone (4MB), 4′-methyoxyacetophenone (4MOA), 6-methyl-5-hepten-2-one (6M5, sulcatone), cyclohexanone (CYC), and indole (IND) (Appendix A). 

Contrary to our expectations, olfactory responses to the unitary compounds in Blend 26 were not significantly different between callow and mature minors, and several compounds actually elicited slightly, albeit non-significantly, higher responses in mature minors (Two-Way ANOVA, *p*-value for age = 0.1497; odor = 0.0684; age × odor blend = 0.8562; Appendix A). Curiously, in a preliminary examination of the olfactory responses of minor workers of unknown age, we observed that Blend 26 also elicited robust—and indeed the highest—responses in nurses when compared with foragers, which were largely indifferent to this blend (Appendix A). Therefore, to distinguish between age- and task-associated differences in olfactory sensitivity, the responses of mature nurses (mean ± SEM = 34.7 ± 3.2 days post-eclosion) and mature foragers (27.7 ± 5.2 days post-eclosion) to the unitary ketones and indoles in Blend 26 were examined. Here, nurses were significantly more responsive than foragers to a sole component of Blend 26—3MI (Two-Way ANOVA with Tukey’s correction for multiple comparisons, *p* = 0.0141, N = 7; Figure 4a). Taken together, these results suggest that nurses (which are generally the youngest workers in the colony) may be driving the strong responses we have observed to Blend 26 in both callow workers and nurses, regardless of their age. Importantly, these results suggest there may be a functional, task-associated role for 3MI sensitivity in *C. floridanus* nurses that is independent of age.

To functionally characterize the valence of 3MI, the attraction/aversion responses of whole colonies to filter paper strips containing either solvent (DMSO) or serial dilutions of 3MI were examined by placing those strips directly within *C. floridanus* colonies (Appendix A). In this novel behavioral paradigm, our *C. floridanus* colonies collectively exhibited a dose-dependent avoidance behavior in response to increasing concentrations of 3MI, and there were significantly fewer ants contacting (i.e., walking, standing) the filter paper strips soaked with 10^−1^ M 3MI than solvent controls (One-Way ANOVA with Tukey’s correction for multiple comparisons, *p* = 0.0038, N = 7; Appendix A). Furthermore, the valence of individual minor worker nurses and foragers to 3MI was also examined using a proximity-based bioassay (Appendix A). Here, valence reflects the solvent-normalized distance (in cm) ants spent in proximity of a filter paper soaked in serial dilutions of 3MI, such that positive numbers represent an attraction and negative numbers represent a repellent effect, with “0” reflecting indifference (Appendix A and Figure 4b). Throughout these studies, nurses were located further from the 3MI odor source than from solvent alone at every concentration tested; indeed, they were significantly repelled by 3MI at the 10^−5^ (One Sample *t*-Test, *p* = 0.0006, N = 8) and 10^−1^ M (*p* = 0.0023, N = 5) concentrations (Figure 4b). In contrast, foragers were only significantly repelled by 3MI at the highest concentration tested (*p* = 0.0100) and instead were significantly attracted to 3MI at lower concentrations (*p* = 0.0180) (Figure 4b). Additionally, foragers spent a significant amount of time standing directly on top of the 3MI odor source at the 10^−5^ M concentration, whereas nurses never did (Two-Way ANOVA with Tukey’s correction for multiple comparisons, *p* = 0.0154, N = 8; Appendix A), further suggesting that foragers may be attracted to 3MI at low concentrations.

## 4. Discussion

Age polyethism, where animals sequentially perform different tasks as they age, is a hallmark of social organization in ant colonies [3]. While age is often correlated with worker ant behavior, it is important to appreciate that age is unlikely to be the sole determinant of caste identity or task allocation. Instead, in complex eusocial ant colonies, these processes are likely driven by ever-changing needs of the colony that are often dictated by their local environment [2,14,19]. Genetic, epigenetic, physiological, and systems-level processes all contribute to regulating the tasks performed by individual workers and the colony-level behavioral patterns that emerge from these interactions [9,10,23,24,25,26,27,28]. Colonies are therefore highly adaptive complex social systems, and the ability of workers to dynamically switch tasks in response to colony need has likely contributed to the remarkable success of eusocial ants. By gaining insight into the mechanisms that govern task allocation and division of labor in ants, we can gain a deeper understanding of cooperation, conflict, self-organization, and the emergent properties of social systems beyond these fascinating insects.

As previously reported in *C. floridanus* and other ant species [2,3,15,16], we observed an age-based division of labor in *C. floridanus* minor workers (Appendix A). Minors typically act as nurses when they are young before undergoing an age-associated transition to other tasks such as cleaning, trophallaxis, and eventually foraging. Consistent with previous studies [2], *C. floridanus* callow minors had a smaller repertoire of tasks than mature minors (Appendix A). This aligns with the hypothesis that older mature minors exhibit a greater degree of behavioral plasticity than younger workers. Importantly, the presence of older nurses and younger foragers within the colony suggests that age is likely only a correlative aspect for behavior rather than a determining agent (Appendix A). In contrast, *C. floridanus* majors rarely engaged in routine tasks within the nest and were never observed nursing (Appendix A). While majors are relatively inactive in this regard, there is mounting evidence to suggest that these distinctive larger workers serve as a morphologically and physiologically specialized soldier caste [9,10,11]. Indeed, *C. floridanus* majors appear to be fine-tuned and highly specialized for colony defense; they display enhanced sensitivity toward CHCs and are correspondingly significantly more efficient at fending off and killing non-nestmate opponents that might otherwise threaten the colony [11].

Consistent with the caste- and age-associated differences in the behavioral repertoire of minors and majors (Appendix A) [9,10], we observed significant shifts in odor coding and olfactory sensitivity among workers. Older, more mature minor workers, which represent the most behaviorally active workers in the colony, exhibited significantly higher baseline (raw) responses to odor blends, distinguishing them from callow minors, callow majors, and mature majors (Figure 1 and Figure 2). The olfactory transition from low to high sensitivity observed as minors aged was not detected in majors (Figure 2). Instead, major workers maintained low odorant sensitivity even as they matured. Taken together, as engagement in colony tasks increases, olfactory sensitivity also increases. It is important to note, however, that these EAG measurements do not take into consideration the solvent-normalized response. Indeed, solvent responses were also highest in mature minors compared with other worker groups (Figure 2a,c–f). Therefore, this broad increase in responsiveness among mature minor workers may reflect critical shifts in the development of the olfactory system (e.g., an increase in the number of OSNs) rather than changes in the ability to detect and perceive odorants. 

Normalizing odor blend responses to control for solvent alone resulted in more nuanced differences among aging worker groups. Majors exhibited a range of sub-solvent, inhibitory responses to odor blends, and this was consistent across age bins (Figure 3). Of particular interest was Blend 26. Several of the unitary compounds that comprise this blend have been previously linked to ant biology and may therefore be biologically salient and functionally significant to *C. floridanus*. In associative learning tests, *C. aethiops* workers tended to generalize (i.e., respond to stimuli that has some similarity to the conditioned stimulus) toward 2OCT regardless of the chemical used for the conditioned stimulus, suggesting that 2OCT may have innate significance [43]. Furthermore, both 2OCT and 3OCT have been described as components of the alarm pheromone in several *Atta* species and *C. schaefferi*, respectively [44,45,46]. Sulcatone (6M5) has also been described as an alarm pheromone in several ants [47,48] and has been shown to elicit strong responses from basiconic sensilla of *C. floridanus* [49]. This blend of ketones and indoles elicited the highest response in callow minors and the highest response to any odor blend relative to other worker groups (Figure 3). Furthermore, nurses were also more sensitive than foragers to this odor blend (Appendix A). That said, there was no significant difference in the responses of callow minors and mature minors to any of the unitary compounds that comprise this blend (Appendix A). Therefore, to disentangle the effects of age and task, we also examined the olfactory responses of mature nurses and relatively younger mature foragers to the components of Blend 26. Here, we found that older mature nurses were significantly more responsive to 3MI, commonly known as skatole, compared to younger mature foragers (Figure 4A). We found that at the colony level, high concentrations of 3MI acted as a repellent (Appendix A). In addition, 3MI elicited significantly more aversive behavior in nurses than in foragers (Figure 4b and Appendix A). Indeed, nurses were significantly repelled by 3MI at both low and high concentrations, whereas foragers showed a significant attraction to 3MI at low concentrations and only manifest a repellent aversion at high concentrations (Figure 4b). 

To our knowledge, a biological role in nature for 3MI has not been specifically described for *C. floridanus*. However, 3MI has been reported to give army ants a distinct fecal odor [50], and it is possible that this compound is a pheromone secreted by one of the many glands found throughout the ant body [3]. In orchid bees and gravid mosquitoes, exogenous 3MI found in the environment serves as an attractant [51,52]. Therefore, it is also possible this compound may be encountered in *C. floridanus*’s local environment, either in their wooded nesting sites or brought to the nest by returning foragers. At any rate, these results suggest that 3MI may be an important odor cue involved in brood care, perhaps as a noxious substance best avoided by the stationary, developing brood that cannot retreat from waste piles or other unfavorable nest conditions without the active support of nurses. Taken together, differences in the sensitivity to a given odorant among workers of a similar age may be responsible for phenotypic differences in behavior, suggesting that variation in olfactory sensitivity may play an important role in task allocation in ant colonies.

## 5. Conclusions

In *C. floridanus*, minor workers play a dynamic role performing most tasks required by the colony. Importantly, minors also manifest profound shifts in olfactory sensitivity that correlate with both age and task which may reflect developmental and behavioral shifts. In contrast, majors maintained consistently low responses to odor blends over time, an observation that aligns with their more restricted and specialized role as soldiers. The identification of 3MI, which elicits significantly different olfactory sensitivity and behavioral valence in nurses and foragers, suggests that this odor cue may play a functional role in brood care. Taken together, these data support a model in which the dynamic regulation of olfactory sensitivity is tightly associated with worker behavior and may therefore serve as an important physiological mechanism governing task allocation in eusocial ant colonies.

## Figures and Tables

**Figure 1 insects-14-00724-f001:**
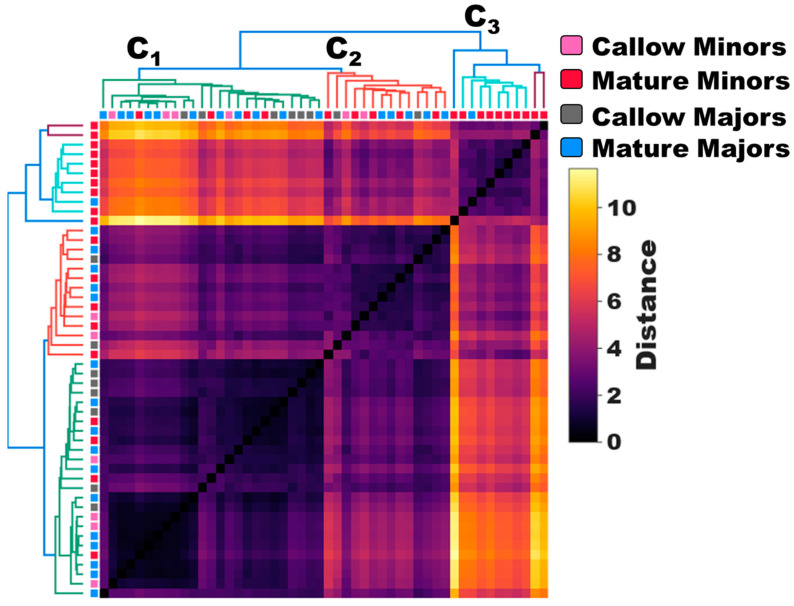
Dendrogram of untransformed EAG responses (mV) of minors and majors. A dendrogram showing hierarchical clustering based on the untransformed EAG responses (mV) of minor and major workers (pink = callow minors; red = mature minors; gray = callow majors; blue = mature majors). N = 25 minors; N = 25 majors.

**Figure 2 insects-14-00724-f002:**
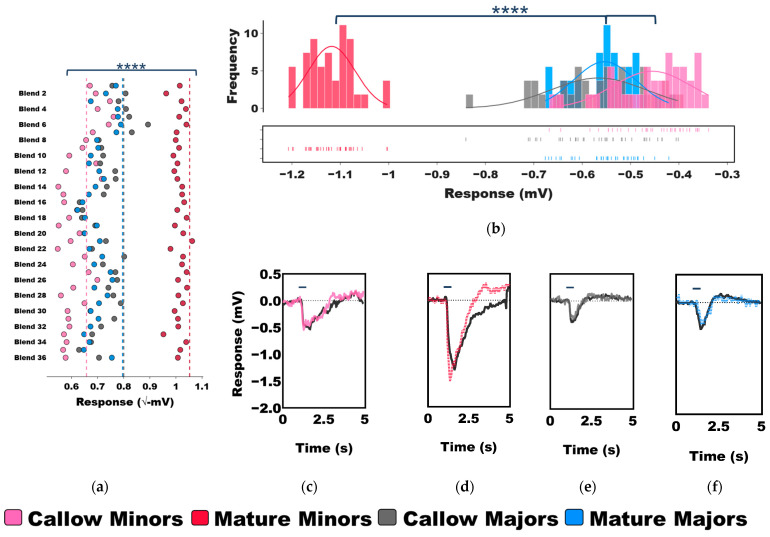
Age-associated shifts in olfactory sensitivity of minors and majors. Active and behaviorally plastic mature minors show increased sensitivity to odor blends compared to other workers within the colony. (**a**–**b**) A dot plot and frequency histogram showing the mean EAG responses (√−mV) of callow minors (pink), mature minors (red), callow majors (gray), and mature majors (blue) to 36 odor blends. The dotted lines show the mean response to solvent (ND96). Dot plot: Two-Way ANOVA, *p*-value for caste < 0.0001 (****); odor blend = 0.7459 (ns); caste × odor blend > 0.9999 (ns). Frequency histogram: One-Way ANOVA with Tukey’s multiple comparison test, **** < 0.0001. N = 6 callow minors; N = 19 mature minors; N = 8 callow majors; N = 17 mature majors. (**c**–**f**) Representative EAG traces of callow minors (pink), mature minors (red), callow majors (gray), and mature majors (blue) to Blend 17. The black line depicts the mean response to solvent for the recording.

**Figure 3 insects-14-00724-f003:**
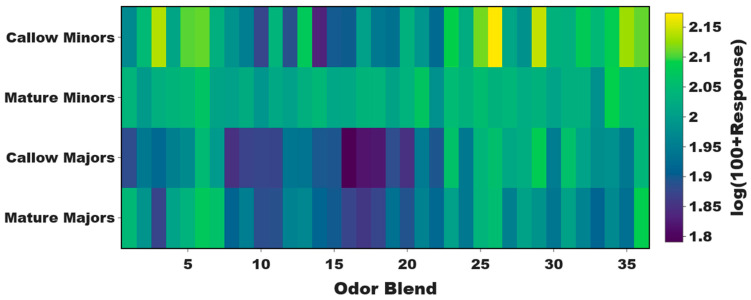
Normalized olfactory responses in aging minors and majors. A heatmap of the mean, solvent (ND96) normalized EAG responses of minors and majors. Two-Way ANOVA, *p*-value for caste < 0.0001; odor blend < 0.0001; caste×odor blend = 0.1426. N = 6 callow minors; N = 19 mature minors; N = 8 callow majors; N = 17 mature majors.

**Figure 4 insects-14-00724-f004:**
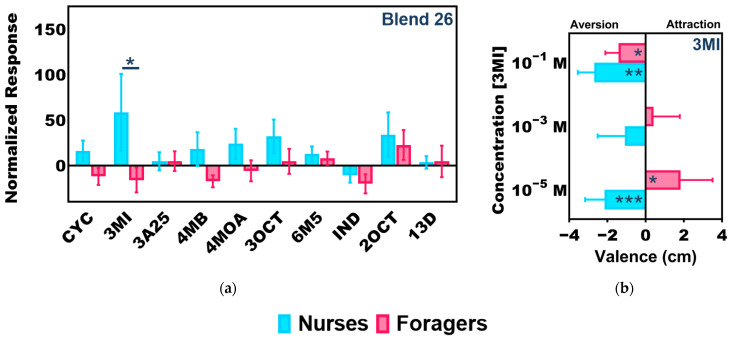
Nurses exhibit significantly higher olfactory sensitivity and aversion to 3MI compared to foragers. Significant olfactory differences between older nurses (mean ± SEM = 34.7 ± 3.2 days post-eclosion) and younger foragers (27.7 ± 5.2 days post-eclosion). (**a**) A bar graph of the mean solvent (ND96) normalized EAG responses of minor nurses (blue) and foragers (red) to the unitary compounds found in odor Blend 26: 1,3-diphenylacetone (13D), 2-octanone (2OCT), 3-acetyl-2,5-dimethylfuran (3A25), 3-methylindole (3MI), 3-octanone (3OCT), 4-(4-methoxyphenyl)-2-butanone (4MB), 4′-methyoxyacetophenone (4MOA), 6-methyl-5-hepten-2-one (6M5), cyclohexanone (CYC), and indole (IND). Two-Way ANOVA with Tukey’s correction for multiple comparisons, *p* = 0.0141 (*), N = 7 nurses; N = 7 foragers. (**b**) Valence as measured by the mean distance (cm) between individual nurses (blue) and foragers (red) and the Whatman paper “zone” containing serial dilutions of 3MI normalized to solvent-alone controls where 0 represents the position of individual ants with respect to solvent-alone, positive numbers indicate attraction, and negative numbers indicate aversion. One Sample *t*-Test, *p*-values: nurses 10^−5^ M = 0.0006 (***); foragers 10^−5^ M = 0.0180 (*); nurses 10^−1^ M = 0.0023 (**); foragers 10^−1^ M = 0.0100 (*). N = 5 × 10^−3^ M, 10^−1^ M; N = 8 × 10^−5^ M.

## Data Availability

All data generated or analyzed during this study are included in this published article and its Appendix A. Raw data have been made available in Appendix A.

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
