# Peer review of "Age and Task Modulate Olfactory Sensitivity in the Florida Carpenter Ant Camponotus floridanus"

_insects, 2023, doi:10.3390/insects14090724_

Round 1

Reviewer 1 Report

The authors of Ferguson et al. reported that the olfactory sensitivity in the Florida carpenter ant Camponotus floridanus was varied in age and task-dependent. They found the younger minor nurses exhibited higher sensitivity to odorants than the older minor foragers, while the mature worker showed higher responses to odorants than the callow workers. The majors remained the consistent response regardless of age. Such changes in olfactory sensitivity might mediate the social behavior regulation in ants. The findings are interesting for addressing the modulatory mechanism of social behavior by environments. The manuscript was well-organized and written, and the work was solid. I have no other comments on the manuscript.

Author Response

The authors of Ferguson et al. reported that the olfactory sensitivity in the Florida carpenter ant Camponotus floridanus was varied in age and task-dependent. They found the younger minor nurses exhibited higher sensitivity to odorants than the older minor foragers, while the mature worker showed higher responses to odorants than the callow workers. The majors remained the consistent response regardless of age. Such changes in olfactory sensitivity might mediate the social behavior regulation in ants. The findings are interesting for addressing the modulatory mechanism of social behavior by environments. The manuscript was well-organized and written, and the work was solid. I have no other comments on the manuscript. We thank the reviewer for their generous comments and their efforts.

Reviewer 2 Report

The study by Ferguson et al. examined the olfactory responses of age and task groups in a carpenter ant Camponotus floridanus, and the authors reported an age-dependent shift in olfactory sensitivity among worker-like minors, but this change was not observed in the soldier-like majors. This is an interesting paper addressing olfactory plasticity associated with the division of labor in eusocial colonies. Overall, the experiments are well designed, and the manuscript is well written. The EAG results were not significantly different between callow and mature workers when transformed to solvent control, but the authors provided an explanation for the solvent effect, and there is likely a broad increase in sensitivity as minor workers age.

There are a few clarifications needed to improve the paper:

Line 169: Some details are missing for this experiment. How many ants were present in each tested colony? The treated filter paper pieces were randomly placed in the colony – were the ants also randomly distributed in the arena to justify this (to rule out a location effect)? What was the size of the arena? Also, it is unclear how the sample size N = 7 (Figure S1) was obtained, as there were only six lab colonies available (line 132).

Line 204: ANOVA and t-tests were used for data analyses, but the manuscript did not describe if the data were tested for normality and equal variance as assumptions of parametric analysis.

Lines 215-226: The description of methods needs be moved to “Materials and Methods” section.

Lines 259-266: Again, description of the odors tested and the rationale can be moved to “Materials and Methods”.

Line 464: The biological relevance of 3-methylindole is still unclear in C. floridanus, and it is unknown if callow minors encounter this odor naturally. Although the function of this compound in other insects is discussed, a little more discussion of its potential sources in C. floridanus (future directions) would be helpful (e.g., also a fecal odor as in the army ants, brought to the nest by foragers from the environment).

Author Response

The study by Ferguson et al. examined the olfactory responses of age and task
groups in a carpenter ant Camponotus floridanus, and the authors reported an age-dependent shift in olfactory sensitivity among worker-like minors, but this change was not observed in the soldierlike majors. This is an interesting paper addressing olfactory plasticity associated with the division of labor in eusocial colonies. Overall, the experiments are well designed, and the manuscript is
well written. The EAG results were not significantly different between callow and mature workers when transformed to solvent control, but the authors provided an explanation for the solvent effect, and there is likely a broad increase in sensitivity as minor workers age.

There are a few clarifications needed to improve the paper:

Line 169: Some details are missing for this experiment. How many ants were present in each tested colony? Our colonies are comprised of several thousand workers (approx. 2000-5000), a single queen, and brood. This information has been added to the revised text. The treated filter paper pieces were randomly placed in the colony – were the ants also randomly distributed in the arena to justify this (to rule out a location effect)? We were observing the response of full,
queen-right colonies, and to that end, the ants were self-organized within the colony. In order to control for positional effects, we randomized the location of the filter paper each trial as opposed to randomizing the position of the ants (which was not possible). What was the size of the arena? 40.6 x 26.7 cm. This information has been added to the revised text. Also, it is unclear how the sample size N = 7 (Figure S1) was obtained, as there were only six lab colonies available (line 132). Information regarding how the colonies were used in this
experiment can be found in the Supplementary File 2.

Line 204: ANOVA and t-tests were used for data analyses, but the manuscript did not describe if the data were tested for normality and equal variance as assumptions of parametric analysis. We have examined the QQ and residual plots for all data. The data for Figure 2A and Figure 3 were subsequently transformed to better satisfy the assumptions of normality and equal variance. While notably, this did not meaningfully change our interpretation of the data, we have included updated figures and statistical results in this revised submission in order to reflect these changes. We have also included additional information in the Materials and Methods in the revised manuscript that address this concern.

Lines 215-226: The description of methods needs be moved to “Materials and Methods” section. We have removed this text from the Results section and note that this information was already available in the Materials and Methods section.

Lines 259-266: Again, description of the odors tested and the rationale can be moved to “Materials and Methods”. This information has been shifted to the Materials and Methods section of our revised manuscript.

Line 464: The biological relevance of 3-methylindole is still unclear in C. floridanus, and it is unknown if callow minors encounter this odor naturally. Although the function of this compound in other insects is discussed, a little more discussion of its potential sources in C. floridanus (future directions) would be helpful (e.g., also a fecal odor as in the army ants, brought to the nest by
foragers from the environment). More information has been added to the revised manuscript to suggest possible sources where C. floridanus may encounter 3-methylindole in nature.

Reviewer 3 Report

This research investigated the effects of age and task group on olfactory responses to a series of odorant blends in minor and major worker castes. They found that significant age-associated shifts in the olfactory responses of minors as they transitioned between behavioral states, whereas the responses of majors remained consistently low regardless of age. Moreover, a compound, 3-methylindole was also identified to play an important role in brood care. With these results, age- and task-associated shifts in olfactory physiology may play a critical role in the social organization of ant colonies. However, some points should be addressed before publication.

Major

1.      The CAS number of chemicals used in this study should be supplementary.

2.      L170-179: what was the solvent used in the experiment of Figure S1? DMSO or ND96? Relative information in the text and figure legend was contradictory. The number of ants tested should be replenished.

3.      The results sections should be modified since it included some information that may be more appropriate in introduction, material and method, and discussion section. L212-225, 233-234, 239-243, 330-337, etc.

4.      The discussion should be modified with more supporting literature but not only the summary of results.

5.      The description of Figure 4 should be clearly correlated with the experimental methods described in the paper.

6.      Materials and method section should be modified and supplemented more information to make it repeatable. e.g. L225, what is the monitor period? For majors (N=16), and minor (N=196), a big gap was obvious in the number of insects used. There was a concern here whether the number of insects used in this study was adequate.

Minor

1.        L106-109: The abbreviations were unnecessary since they were mentioned anywhere below.

2.        L144, L154-168, L382, L408-409, etc.: The superscript and subscript of some unit, chemical compounds should be formalized. The manuscript should be fully checked to avoid similar errors.

3.        L246-247: the terms 'younger callow workers (<6 days post-eclosion)' and 'older mature workers (≥6 days post-eclosion)' should be consistently used in the subsequent figures and throughout the article for the ease of readers' understanding and reference. Additionally, please include four types of categorization: 'callow minors, mature minors, callow majors, and mature majors.'"

4.        The figure legend in the text should be lowercase.

Author Response

This research investigated the effects of age and task group on olfactory responses to a series of odorant blends in minor and major worker castes. They found that significant ageassociated shifts in the olfactory responses of minors as they transitioned between behavioral states, whereas the responses of majors remained consistently low regardless of age. Moreover, a compound, 3-methylindole was also identified to play an important role in brood care. With these results, age- and task-associated shifts in olfactory physiology may play a critical role in the social organization of ant colonies. However, some points should be addressed before publication.

Major

1. The CAS number of chemicals used in this study should be supplementary. The revised Supplementary File 1 now has the CAS numbers for each chemical used in this study.

2. L170-179: what was the solvent used in the experiment of Figure S1? DMSO or ND96? Relative information in the text and figure legend was contradictory. This contradiction has been corrected in the revised figure legend. The number of ants tested should be replenished. The number of ants per colony has been added to the caption (as well as to the Materials and Methods).

3. The results sections should be modified since it included some information that may be more appropriate in introduction, material and method, and discussion section. L212-225, 233-234, 239-243, 330-337, etc. We agree. These sections have been shifted to the more appropriate sections of the manuscript or, when appropriate we have removed redundant information.

4. The discussion should be modified with more supporting literature but not only the summary of results. In addition to shifting additional text and citations to the Discussion section (as discussed above), we have also included an enhanced text as well as additional supporting literature which indeed, was critically missing from the Discussion section.

5. The description of Figure 4 should be clearly correlated with the experimental methods described in the paper. A description of the methods used for the EAG data presented in Figure 4 can be found in Section 2.3 of the Materials and Methods, entitled ‘Electroantennography’. A description of the methods used for the behavioral valence data in Figure 4 can be found in section 2.5. of the Materials and Methods, entitled ‘Individual Responses to 3-Methylindole’.

6. Materials and method section should be modified and supplemented more information to make it repeatable. e.g. L225, what is the monitor period? For majors (N=16), and minor (N=196), a big gap was obvious in the number of insects used. There was a concern here whether the number of insects used in this study was adequate. Minors and majors were painted in approximately equal numbers. The N number refers to the total number of observations made of ants engaged in pre-specified social behaviors. Therefore, we observed more painted minors engaged in nursing, foraging, trophallaxis, and cleaning behaviors compared to painted majors. We posit that the gap in these numbers reflects
the distinctly different roles of minors and majors. This information can be found in section 2.2 of the Materials and Methods, entitled ‘Paint marking, behavioral monitoring, and collections’.

Minor

1. L106-109: The abbreviations were unnecessary since they were mentioned anywhere below. We agree and have removed the abbreviations for ionotropic receptors (IRs) and gustatory receptors (GRs).

2. L144, L154-168, L382, L408-409, etc.: The superscript and subscript of some unit, chemical compounds should be formalized. The manuscript should be fully checked to avoid similar errors. We have corrected these and other inadvertent errors encountered throughout the text.

3. L246-247: the terms 'younger callow workers (<6 days post-eclosion)' and 'older mature workers (≥6 days post-eclosion)' should be consistently used in the subsequent figures and throughout the article for the ease of readers' understanding and reference. Additionally, please include four types of categorization: 'callow minors, mature minors, callow majors, and mature majors.'" The text that introduces these categorizations has been revised to
emphasize the four different categorization types, and we have provided additional clarity throughout the text where appropriate.

4. The figure legend in the text should be lowercase. With respect, we believe that all figure legends in the text are using the appropriate sentence case.

Round 2

Reviewer 3 Report

The authors have addressed my points and I would suggest it to be accepted.